# A Review of Battery Equalizer Circuits for Electric Vehicle Applications

**Alfredo Alvarez-Diazcomas** [1,†] , **Adyr A. Estévez-Bén** [1,2,†] , **Juvenal Rodríguez-Reséndiz** [1,*,†] , **Miguel-Angel Martínez-Prado** [1,†] , **Roberto V. Carrillo-Serrano** [1,†] and **Suresh Thenozhi** [1,†]

1   Facultad de Ingeniería, Universidad Autónoma de Querétaro, Querétaro 76010, Mexico; aalvarez78@alumnos.uaq.mx (A.A.-D.); aestevez05@alumnos.uaq.mx (A.A.E.-B.); miguel.prado@uaq.mx (M.-A.M.-P.); roberto.carrillo@uaq.mx (R.V.C.-S.); suresh@uaq.mx (S.T.)
2   Facultad de Química, Universidad Autónoma de Querétaro, Querétaro 76010, Mexico
*   Correspondence: juvenal@uaq.edu.mx; Tel.: +52-442-192-1200
†   These authors contributed equally to this work.

**Abstract:** Electric vehicles (EVs) are an alternative to internal combustion engine (ICE) cars, as they can reduce the environmental impact of transportation. The bottleneck for EVs is the high-voltage battery pack, which utilizes most of the space and increases the weight of the vehicle. Currently, the main challenge for the electronics industry is the cell equalization of the battery pack. This paper gives an overview of the research works related to battery equalizer circuits (BECs) used in EV applications. Several simulations were carried out for the main BEC topologies with the same initial conditions. The results obtained were used to perform a quantitative analysis between these schemes. Moreover, this review highlights important issues, challenges, variables and parameters associated with the battery pack equalizers and provides recommendations for future investigations. We think that this work will lead to an increase in efforts on the development of an advanced BEC for EV applications.

**Keywords:** lithium-ion battery; battery management systems; battery equalizer circuits; electric vehicles

## 1. Introduction

Global warming is one of the biggest challenges today for humankind. The increase in temperature has caused the disappearance of animal and plant species, defrosting of glaciers, sea level rise, extreme weather events and many other phenomena that threaten life on our planet as we know it. The main cause of this change is the emission of greenhouse gases into the atmosphere. These gases allow the light coming from the sun to pass through them and reach Earth. However, they keep part of the radiation that is bounced back from the surface of the Earth [1–4].

Some of the leading sources of these greenhouse gases are electricity generation, transportation, industry, agriculture, and the commercial and residential sectors. The transportation sector is one of the most significant contributors, representing 23.96% of total emissions of $CO_2$ worldwide [5]. Moreover, it is responsible for the higher growth in emissions today due to the growth of tourism, the globalized economy and the increase in living standards [5,6].

A viable alternative to reduce emissions in this sector is the use of EVs, which practically behave like zero-emission cars. Despite the recent interest in these automobiles, their invention dates back to the nineteenth century. William Morrison built the first successful electric car in the United States of America (USA) in 1891. By 1914, the sales of these cars began an irreversible and inevitable decline due to competition with ICE automobiles. They never disappeared completely, but were limited to light-duty vehicles [7,8].

Most reasons why these cars never had extensive use remain today. One of the main obstacles is the autonomy of the car since it depends on the battery. In addition, the charging time makes it unattractive, they have a high selling price, and there is not a large number of charging stations. However, currently, they present a comparable performance to ICE-based vehicles [9,10].

Despite the above mentioned limitations, benefits have been provided in the USA to encourage the purchase of these cars due to their positive environmental impact. Some examples are credits for purchase, access to shared travel lanes, exemption from inspections, and reduction of registration fees, among others [11,12]. These and other factors have caused the growth of sales of these vehicles by seven times from 2010 to 2015 [13–15].

New challenges have emerged in the electronic industry for EVs application with the accelerated increase in sales of these automobiles. In [16–18], the main standout trends of the research applied to these cars are described as follows: improving and decreasing the size of the battery chargers from the grid, creation of DC–DC converters for the interface of the sources with a DC bus and the creation of new inverter topologies for the traction system. The main issue related to the battery identified in these papers is the cell equalization.

Typically, an EV battery pack consists of a cluster of cells, where each Li-ion cell is not exactly equal to the others in terms of capacity, internal resistance and self-discharge rate because of normal dispersion during manufacturing. These characteristics cause a different charge/discharge time for each cell, which can lead to the undercharge, overcharge or over-discharge on some cells if the battery pack is operated without protection [16,19]. In these states, the cell loses capacity and can explode; consequently, avoiding them is desirable. The most viable solution for this problem is not found by modifying the chemistry of the battery, but it is found in the electronic industry. Hence, the battery pack is equipped with cell equalizers to avoid the states mentioned above [19–21].

A BEC is essentially a power electronic controller, which takes active measures to equalize the voltage or the state of charge (SOC) in each cell [22–24]. As a result, each of the cells has the same SOC during charging and discharging, even in conditions of high dispersion in capacity and internal resistance. If all the cells have the same SOC utilization, they degrade equally at the average degradation of the pack. If this condition is accomplished, then all cells have the same capacity during the whole lifetime of the battery pack, avoiding premature end of life due to the end of life of only one cell [25–27]. A diagram of these devices is presented in Figure 1.

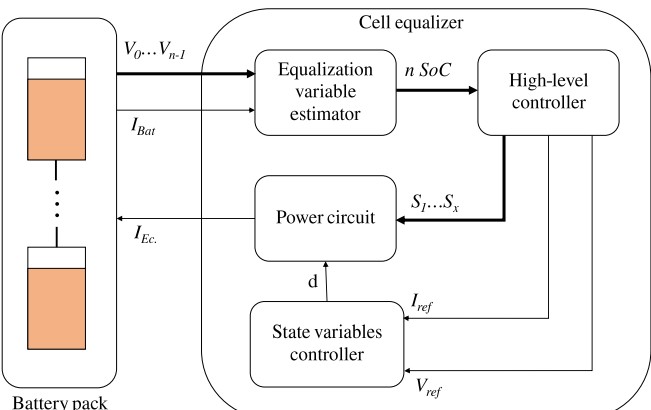

**Figure 1.** General diagram of a cell equalizer.

There are several variables used to decide the homogeneity of the battery pack. The operating voltage of the cell is widely used because it is pretty straightforward to understand and the tension is directly measured. However, this variable does not reflect the internal state of the cell and it is affected by many internal parameters that yield in fluctuations of the voltage and the activation of the equalization process [28–31]. If the operating voltage were used, the equalization variable estimator is not used. When the equalization variable is the SOC or the capacity of the cell, these variables

are not measured directly and require a state estimator. Compared with the operating voltage, these methods reflect the internal state of the battery more accurately and present a lower equalization time. Moreover, it is not affected by the aging process and makes full use of the power of the battery pack. Nevertheless, the main drawback of this variable is its complexity to be obtained accurately. Therefore, the design time is increased and it requires a powerful hardware to be implemented [32–35]. It is well known that batteries are indeed the main hurdle to driving EVs and, as stated above, the main issue for the electronics industry is the cell equalization [16,36]. There are several papers in the literature that present a review of BEC and make a qualitative analysis of these devices [37–40]. However, in this work, the simulation for the main BECs was performed to reach conclusions and make a quantitative analysis. We think that this article will lead to further investigations associated with BEC for EVs application. In Section 2, the power topologies to achieve the equalization are reviewed. In Section 3, the main ideas of this work are discussed and future research opportunities are presented. Finally, in the Conclusions, the main generalizations are summarized and a critical point of view of the authors about the topic is included.

## 2. Battery Equalizer Circuits Applied to EVS

The power circuit is an important subsystem to achieve cell equalization. Many equalization circuits have been presented in the literature, as shown in Figure 2 [38,39]. BECs present the electronics to extract energy from one cell and to transfer that energy to another cell. A high-level controller uses that function to keep the SOC homogeneous across the battery bank. BECs are categorized in a passive or active equalizer depending on if it dissipates energy [39,40]. Moreover, based on the main element of the BEC, BECs are classified as depicted in Figure 2. The frame color of the box indicates the possible transfers of energy. Table 1 explains the color code used in Figure 2.

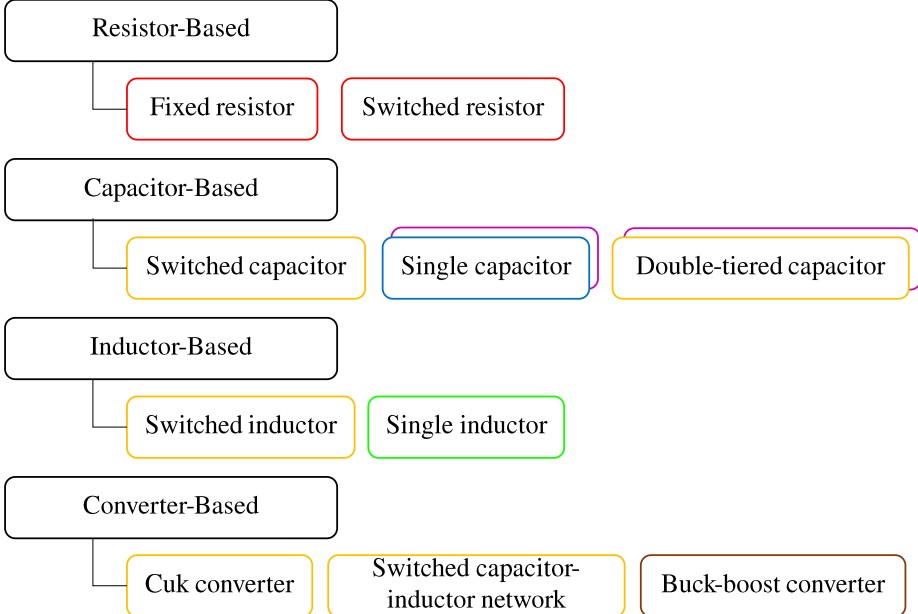

**Figure 2.** Element-based classification of battery equalizer circuits (BECs) [40].

**Table 1.** Figure legend.

| Transfer | Frame Color | Discussion |
|---|---|---|
| Cell-to-heat | Red | Implies the use of a resistor where is burned the excess of energy |
| Adjacent cell-to-cell | Yellow | The BEC can only exchange energy between adjacent cells. |
| Direct cell-to-cell | Blue | The BEC can exchange energy between any cells in the battery pack. |
| String-to-string | Fuchsia | The BEC can exchange energy between arrays of cells. |
| Pack-to-cell | - | The BEC extracts the energy from the whole battery pack and sends it to one cell. |
| Cell-to-pack | - | The BEC extract the energy from an individual cell and send it to the battery pack. |
| String-to-cell | Brown | The BEC extract the energy from an array of cells and send it to one cell. |
| Cell-to-string | Brown | The BEC extracts the energy from an individual cell and sends it to an array of cells. |
| All of the above | Green | The BEC can perform any of the above methods for energy transfer. |

## 2.1. Passive Methods

Passive BECs keep the operating voltage of the cells by burning the excess of energy. Figure 3 shows the switched resistor topology. It can be appreciated that these equalizers present a switch and a resistor for each cell. In this scheme, the metal–oxide–semiconductor field-effect transistor (MOSFET) controls the amount of energy burned. The resistor transforms energy into heat when the MOSFET is on. The switch is on until the cell reaches the lowest voltage in the pack. Therefore, avoiding damaged cells is crucial to prevent an excess of wasted energy. These equalizers present a low efficiency since the goal is to burn energy. Moreover, they require a thermal management system [41–43]. Figure 4 shows the control circuit required for this scheme, where $V_i$ is the voltage of the $i$th cell of the battery pack, $V_{LV}$ is the lowest voltage cell in the battery pack, and $S_i$ is the control signal of the $i$th MOSFET.

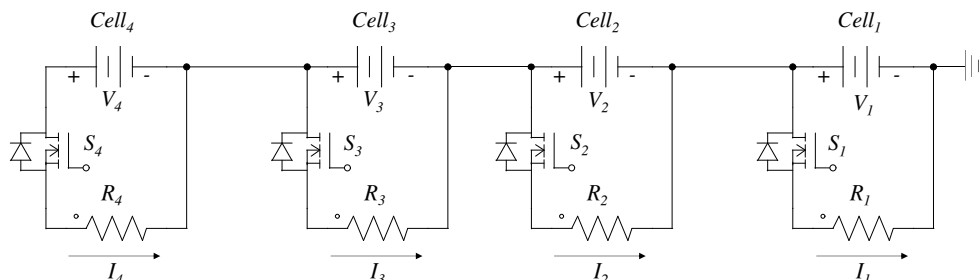

**Figure 3.** Switched resistor equalizer scheme.

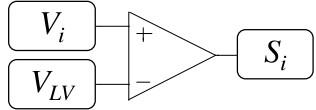

**Figure 4.** Control circuit of the switched resistor topology.

Figure 5 shows the simulation of the equalization process of a four-cell battery bank using the switched resistor topology. The resistance presents a value of 1 Ω, leading to a current numerically equal to the voltage of the cell (2.7–4.2 A). The maximum power dissipation required is the square of the maximum current (17.64 W). It can be appreciated that the energy burned is 200 mAh, 150 mAh

and 100 mAh in cells 2–4, respectively. The equalization time depends on the amount of energy required to burn and the voltage of the cell. Equation (1) describes the behavior of the equalization time in this scheme, where $t_e$ is the equalization time, $Q_{exceeded}$ is the excess of energy in one cell compared to the cell with the lowest SOC and $I_{eq}$ is the equalization current. Hence, the equalization time is inversely proportional to the voltage of the cell. Moreover, activating the equalization process when the cells present the highest possible voltage is desirable. The excess energy is always related to the cell with lower SOC; hence, avoiding a damaged cell is imperative since it will lead to excessive energy waste.

$$t_e = \frac{3600 Q_{exceeded}}{I_{eq}(t)} [s] \qquad (1)$$

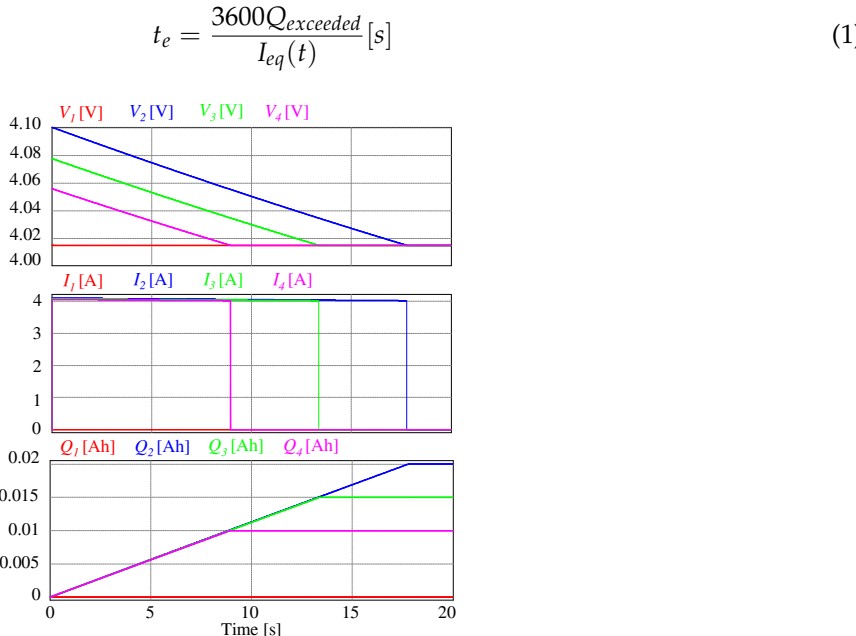

**Figure 5.** Equalization of a four-cell battery bank using the switched resistor scheme.

Reference [44] presents the shunt switch scheme for passive equalization. Previous works have used this topology [42,45]. However, in [44], the MOSFET is used as a voltage-dependent current source achieving a very flexible topology. Figure 6 shows the shunt switch scheme. This scheme only presents one MOSFET per cell. The operating principle is the same as the switched resistor topology; the energy is extracted from each cell and transformed into heat until all cells equal the lowest SOC cell. However, this topology eliminates the resistor and is obtained a reduced component count circuit. Nevertheless, the complexity of the controller required in this scheme is greater than the required in the switched resistor topology. Moreover, the MOSFET must operate in the ohmic region. The desired equalization current is controlled with the voltage gate-source of the MOSFET. Figure 7 shows the algorithm to achieve the desired behavior in the MOSFET. As in the previous topology, avoiding damaged cells in the battery bank is imperative, and the efficiency is a demerit of this scheme. However, it does not require a large thermal management system since the flange of the MOSFET is used.

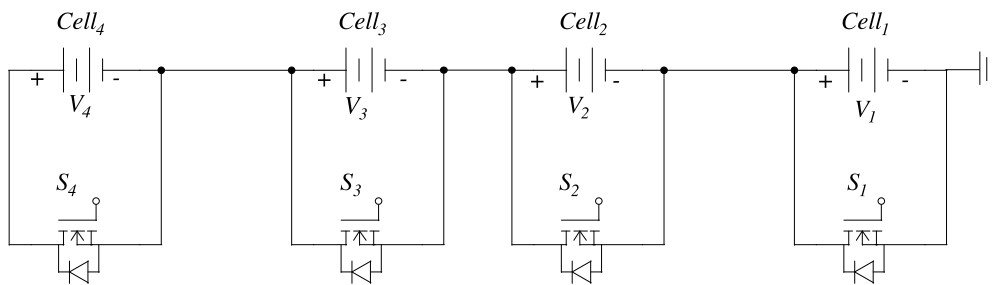

**Figure 6.** Shunt switch equalizer scheme.

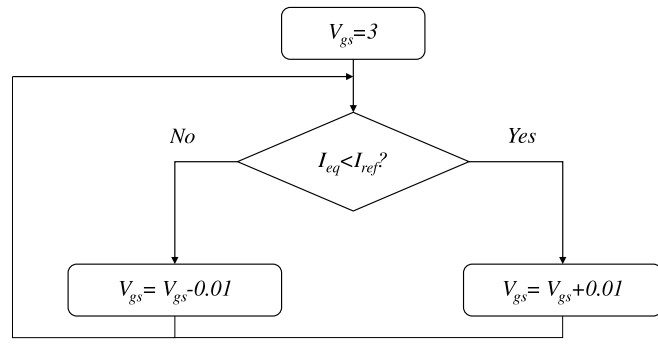

**Figure 7.** Algorithm for the operation of the shunt switch equalizer.

Figure 8 shows the simulation of the equalization process of a four-cell battery bank using the shunt switch circuit. The current is controlled to 4 A with the gate-source voltage of each MOSFET. In this circuit, the current does not depend on the voltage of the cell and the value of a fixed resistor. Hence, the equalization time can be controlled with the reference current. Equation (1) also describes the behavior of the equalization time for this circuit. It can be appreciated that the equalization time was slightly higher with this scheme. However, using the proper reference for the current can emulate the previous behavior or even improve it. The main disadvantage of this configuration is the low efficiency and that it must avoid damaged cells. Table 2 shows a comparison of the passive equalizers.

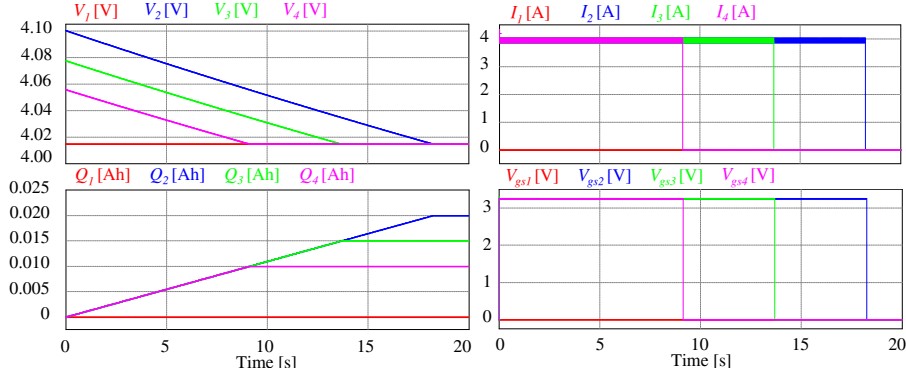

**Figure 8.** Equalization of a four-cell battery bank using the shunt switch scheme.

**Table 2.** Comparative analysis of the passive equalizers.

| Equalizer | Component Count | Equalization Time | MOSFET Stress | Efficiency |
|---|---|---|---|---|
| Switched resistor [40] | *n* resistors *n* MOSFETs | Directly proportional to the energy that wants to be burned and inversely proportional to the voltage of the cell | Voltage of the cell Current reference (ideally the maximum current allowed for the cell) | Poor efficiency since the goal is to burn the excess of energy in the cell |
| Shunt MOSFET [44] | *n* MOSFETs | Directly proportional to the energy that wants to be burned and inversely proportional to the reference current | Voltage of the cell Current reference (ideally the maximum current allowed for the cell) | Poor efficiency since the goal is to burn the excess of energy in the cell |

## 2.2. Active Methods

Active equalizers transfer the excess of energy in one cell to another cell with low SOC. Hence, they present a high efficiency when compared to passive topologies. However, they are costly and complex to control [46]. They are classified considering the main component, as is illustrated in Figure 2 (capacitor-based, inductor-based and converter-based). Moreover, the possible transferences are divided into cell-to-cell (C2C), string-to-cell (S2C), cell-to-string (C2S), pack-to-cell (P2C), cell-to-pack (C2P), string-to-string (S2S) and layer-based [22,40,46].

### 2.2.1. Capacitor-Based Equalizers

The capacitor-based topologies use this element to transfer the energy between cells. Figure 9 shows the switched capacitor equalizer. This circuit is pretty straightforward to understand and to control. It presents a common capacitor for two adjacent cells. The switches connect the capacitor to a cell and its adjacent alternately. A pulse width modulation (PWM) signal applied to the switches causes that behavior. Hence, this is the only control required in this scheme. In one-half cycle of the square wave, the cell with the higher voltage delivers the energy to the capacitor. In the other half of the cycle, the cell receives the energy from the capacitor. The circuit requires two MOSFETs for each cell and one capacitor for every pair of adjacent cells [47].

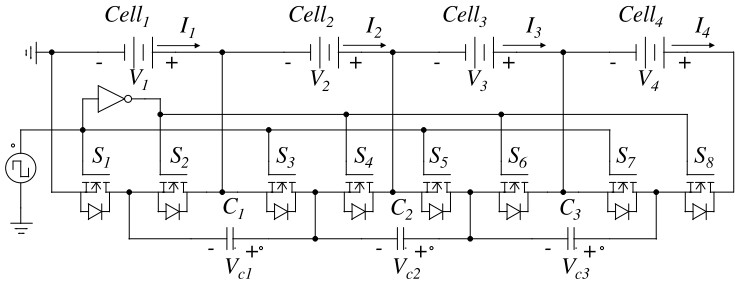

**Figure 9.** Switched capacitor topology equalizer.

The equalization current is not controlled in this scheme. Therefore, it is not possible to avoid surge current in the output of the cell and present a considerable equalization time. Equation (2) shows the behavior of the current in the capacitor $C_1$ of Figure 9, where $i_{C_1}$ is the current in the capacitor $C_1$, $V_x$ is the voltage in the $Cell_x$, $R$ is the resistance of the path of the current and $C_1$ is the capacitance of the capacitor $C_1$. The peak current depends on the difference in voltage between the adjacent cells and the resistance of the current path. This peak current should not surpass a threshold current to avoid damage in the cells. However, it is impossible to avoid dangerous current if the difference in voltage between adjacent cells is large [38,47,48].

Moreover, the factor $RC_1$ is very important since it affects the speed of the system. The current becomes zero after approximately $5RC_1$. After that time, the switch changes its state. In this way, the maximum energy is transferred, and the switching losses are neglected [48]. This work used a capacitor of 47 µF, a resistance of 0.5 Ω to emulate the series resistance of the current path and a switching frequency of 4 kHz.

$$I_{C_1}(t) = \frac{V_1 - V_2}{R} e^{-\frac{t}{RC_1}} \tag{2}$$

Figure 10 shows the simulation for the switched capacitor circuit. It takes 8000 s (more than two hours) to equalize the cells within a range of 10 mV. Moreover, the signal keeps equalizing the cells, and at 16,000 s, all cells are within a hysteresis of 0.004 V. The extensive equalization time is because the current is not controlled, and after a peak in the current, it decreases to zero. Furthermore, the closer the cell voltages get, the lower the peak current became. However, this behavior leads to the controller's simplicity since it does not require a stop condition. Figure 11 depicts a zoom in the variables of interest in the circuit. It shows the behavior of the current and its dependence on the difference in voltage

between adjacent cells. It also shows that the MOSFETs switch the state after the current became zero to maximize the energy transfer and decrease the switching losses. As a demerit, it presents redundant transfers of energy, as exemplified in the transfer between $Cell_2$ and $Cell_3$. In the beginning, $V_2 > V_3$; therefore, $Cell_2$ delivers energy to $Cell_3$. However, in a later stage, the direction of the transfer of energy is reverted. Hence, the process is redundant, and that impacts negatively on the efficiency.

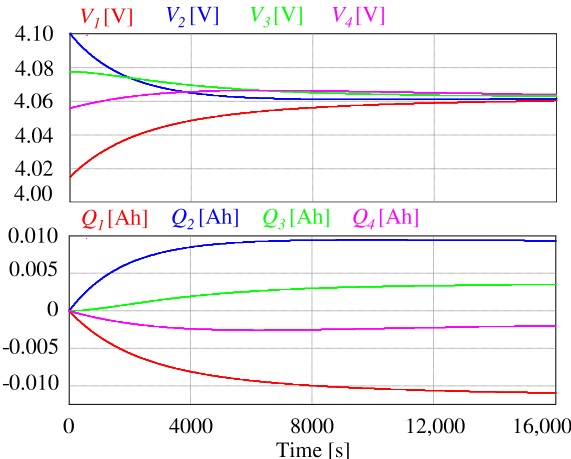

**Figure 10.** Equalization of a four-cell battery bank using the switched capacitor scheme.

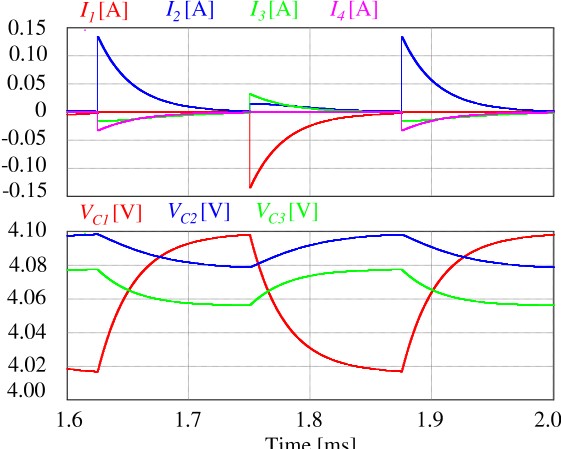

**Figure 11.** Zoom of the current in the cells and voltage of the capacitors in the switched capacitor scheme.

Another capacitor-based topology is the single-capacitor equalizer, depicted in Figure 12. This circuit only requires one capacitor. However, it requires two bidirectional switches per cell (formed with two MOSFET each). The same operating principle explained in the previous topology rules this circuit. The difference lies in the use of only one capacitor [49,50]. Hence, a high-level controller decides the two cells that require equalization. A rule-based controller was implemented for this work following the flowchart illustrated in Figure 13. The controller presents a high complexity in this strategy since the rules provide for the equalization of the cells and the stop condition [51–53].

Figure 14 shows the simulation for the single capacitor equalizer. It is appreciated that the equalization time increases up to 28,000 s (more than seven hours). This increase is because it presents the same characteristics as the previous topology, and only one capacitor to handle all the transfers needed. Therefore, it requires 20,000 s more than the switched capacitor scheme. In general, the single capacitor equalizer presents poor characteristics when compared to the switched capacitor. An advantage is using only one capacitor, and the efficiency is slightly superior because it eliminates redundant transfers of energy. Moreover, it allows direct cell-to-cell transfers, but it requires a complex

controller to take advantage of that feature. This circuit only presents one capacitor but requires one voltage sensor for each cell, double the MOSFETs, and a controller to make decisions.

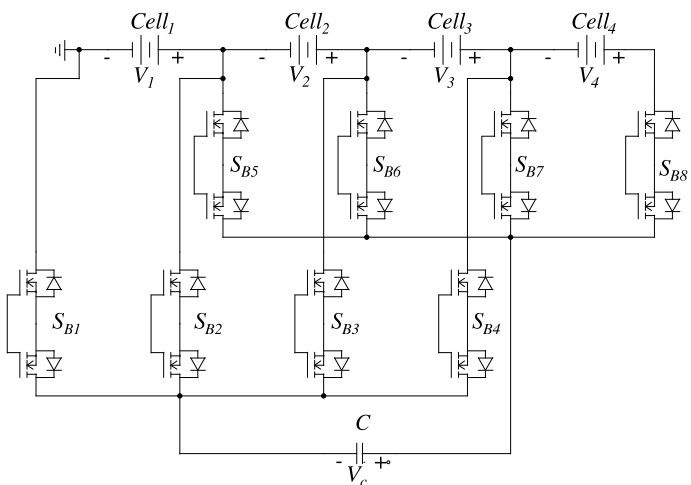

**Figure 12.** Single capacitor topology equalizer.

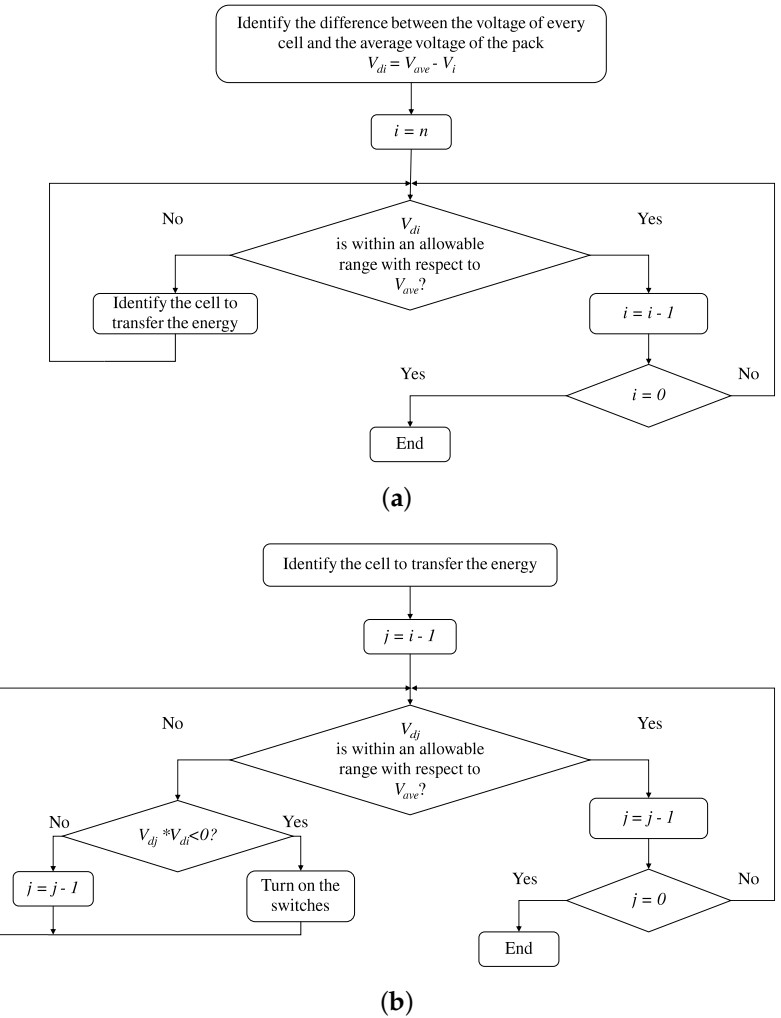

**Figure 13.** Algorithm required to equalize the battery bank: (**a**) general diagram; (**b**) function of the general diagram.

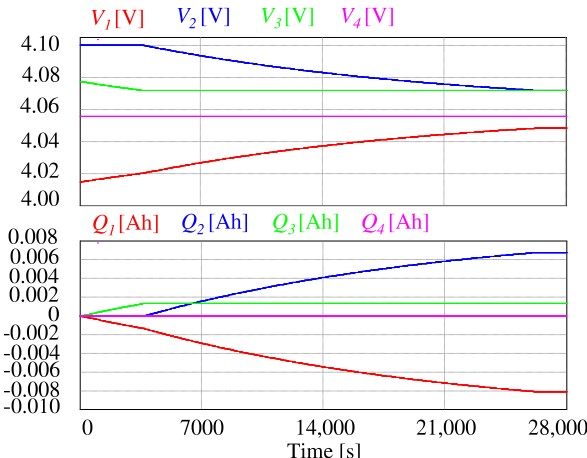

**Figure 14.** Equalization of a four-cell battery bank using the switched capacitor scheme.

Figure 15 shows the double-tiered capacitor equalizer. This scheme is very similar to the switched capacitor; however, a second layer of capacitors is added to speed up the process in this circuit. It presents the same operating principle; hence, the only controller required is a PWM signal. The current is not controlled, and it cannot be avoided a peak current in the switch of the MOSFET. It also presents redundant transfers of energy as the switched capacitor equalizer [54].

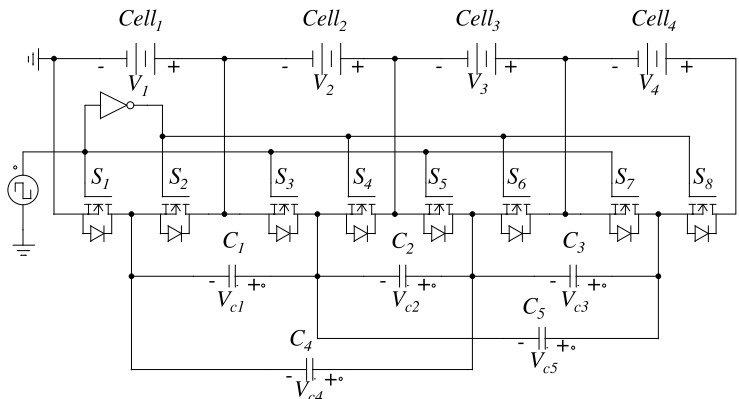

**Figure 15.** Double-tiered capacitor topology equalizer.

Figure 16 shows the simulation for the double-tiered capacitor equalizer. The equalization process takes 12,000 s (more than three hours). The decrease in the equalization time is due to the second layer of capacitors. It is a good value for the capacitors added since the equalization time is decreased a 66.25% when compared to the switched capacitor architecture. Moreover, despite the literature not mentioning it, the same principle can be applied to add more layers of capacitors. In this way, the equalization time is reduced even further. Figure 17 shows a test for a triple-tiered capacitor equalizer. In this case, the mark of 0.04 V is achieved faster (3800 s). Therefore, the designer can decide the use of more layers of capacitors to decrease the equalization time. However, in the triple-tiered switched capacitor scheme, the equalization time is decreased by a 29.63% compared to the double-tiered capacitor equalizer. It can be added layers of capacitors while these elements handle the stress. Table 3 shows a comparison of the capacitor-based equalizers.

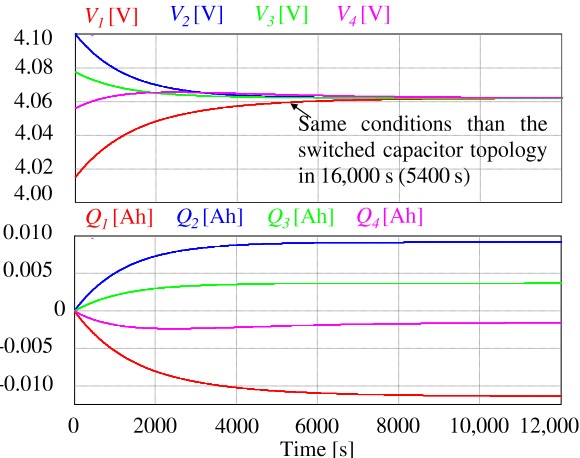

**Figure 16.** Equalization of a four-cell battery bank using the double-tiered switched capacitor scheme.

**Table 3.** Comparative analysis of capacitor-based equalizers.

| Equalizer | Component Count | Equalization Time | MOSFET Stress | Efficiency |
|---|---|---|---|---|
| Switched capacitor [51,52] | $2n$ MOSFETs $n-1$ capacitors | Directly proportional to difference in voltage between adjacent cell and switching frequency, and inversely proportional to the the value of the capacitor and the value of the series resistance. In general, it is a slow process. The equalization time decreases while the amount of capacitors are increased. Hence, the double tiered topology is faster than the switched capacitor and the switched capacitor is faster than the single capacitor topology | Voltage of the cell Peak current ruled by Equation (2) | Only presents conduction losses, if the switching frequency is properly selected. The efficiency is affected negatively by the redundant equalization. |
| Single capacitor [49,50] | $4n$ MOSFETs 1 capacitor | | Voltage of the cell Peak current ruled by Equation (2) | Only presents conduction losses, if the switching frequency is properly selected. It does not present redundant equalization. |
| Double-tiered capacitor [54] | $2n$ MOSFETs $2n-3$ capacitors | | First Layer: Voltage of the cell Second Layer: Two times the voltage of the cell Peak current ruled by Equation (2) | Only presents conduction losses, if the switching frequency is properly selected. The efficiency is affected negatively by the redundant equalization. |

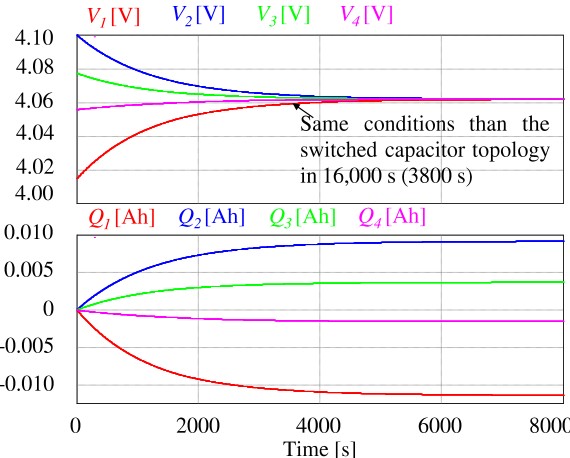

**Figure 17.** Equalization of a four-cell battery bank using the triple-tiered switched capacitor scheme.

### 2.2.2. Equalizers Based on Inductors

There are several schemes in literature that use inductors to transfer energy between cells. These elements allow to control the current extracted from the cells. Therefore, it is possible to protect the cells from potentially dangerous current and reduce the equalization time. Nevertheless, these topologies require complex control techniques, they present magnetizing losses and they are bulky and expensive [38,55,56].

The principle of operation of these topologies is to alternate the connection of the inductor in parallel with the cells of the battery bank. Like capacitor-based equalizers, the inductor stores energy extracted from a cell with a higher SOC in the first stage. This energy is then transferred to another cell with lower SOC to complete the transference. In the first stage, the inductor current increases; in the second stage, it decreases [22,46]. The controller used in this work takes advantage of this operation to control the current. A switch stays on while the current is below the reference and turns off when the current exceeds the reference value. It is necessary to introduce a frequency limiter to avoid significant stress in the switches due to the commutations. Figure 18 shows the circuit designed to achieve the controller operation.

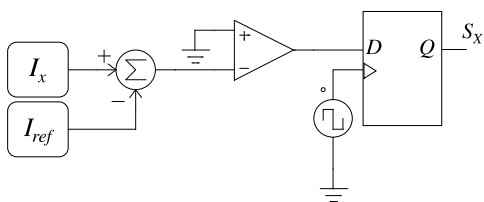

**Figure 18.** Current controller.

Figure 19 depicts the switched inductor equalizer. This scheme transfer energy between adjacent cells. For this purpose, it requires two MOSFETs and one inductor for every pair of neighboring cells. The MOSFETs allow the transfer of energy in any direction. However, it is necessary to act on the appropriate switch depending on the voltage of each cell. For example, if the voltage in $Cell_1$ is greater than the voltage in $Cell_2$, the controller signal must be applied to switch $S_1$, while switch $S_2$ must be off all the time. Hence, when the switch $S_1$ is turned off, the current flows through the antiparallel diode of switch $S_2$ to charge $Cell_2$. In the controller, it is necessary to obtain the measured current modulus to compare it with the reference. Figure 20 shows the controller with the proper modifications for this scheme. Moreover, a high-level controller is required to handle the stop condition (Enable signal in Figure 20). The stop condition depends on the resolution of the sensor [38,55,57].

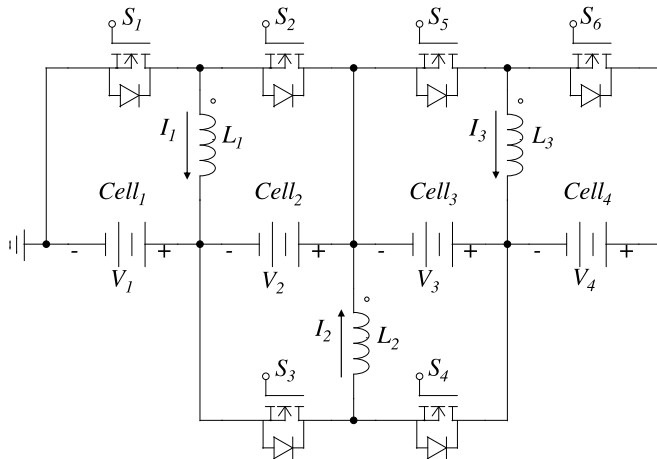

**Figure 19.** Multi-inductor switched equalizer.

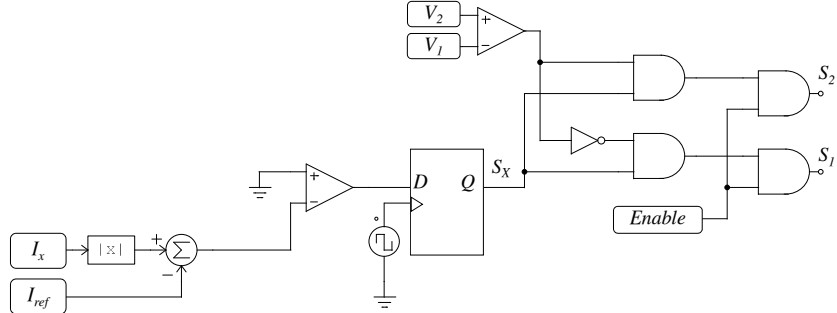

**Figure 20.** Current controller necessary in the switched inductor equalizer.

Figure 21 shows the simulation for a four-cell battery bank with a switched inductor equalizer. This process is faster than the capacitor-based equalizers since it only takes 16 s to finish the equalization. It is comparable to the equalization time obtained with passive equalizers. This characteristic is due to the regulation of the current extracted/delivered from/to the cells. One of the drawbacks of this scheme is that it only features adjacent C2C transfers. Current $I_2$ illustrates this behavior. To transfer energy from $Cell_2$ to $Cell_4$, it must pass for the $Cell_3$ before. In this work, an inductor of 500 µF was used, leading to a ripple of 0.4 A. The value of the inductor leads to a bulky element. The value of the inductance can be decreased to obtain a lighter device. However, this will increase the current ripple. Therefore, it is necessary to find a compromise to achieve desirable characteristics in both parameters.

Figure 22 shows the single inductor equalizer proposed in reference [58]. This topology only requires one inductor to deal with all transfers needed in the battery bank. Therefore, since only one storage device handles all transfers, this scheme requires a larger time to achieve equalization than the switched inductor topology. Besides the inductor, the circuit requires two diodes and two MOSFETs for each cell. This scheme is versatile since C2C, C2S, P2C, C2P and S2S transfers are possible. However, to take advantage of all those possible transfers, a complex high-level controller is required. This high-level controller rules the stop condition, and it sends to the proper MOSFETs the signal $S_x$.

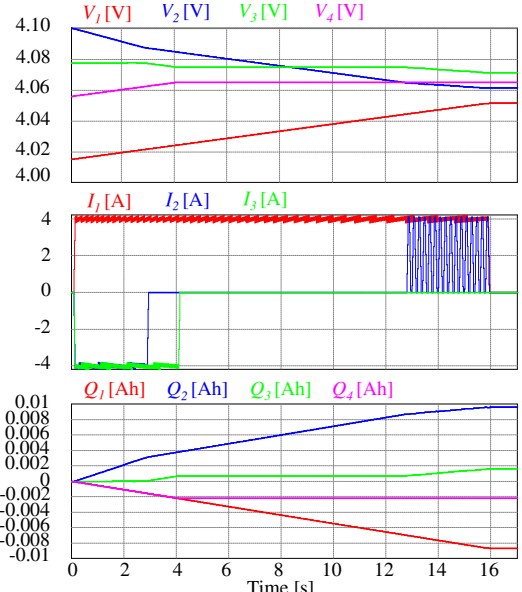

**Figure 21.** Equalization of a four-cell battery bank using the switched inductor scheme.

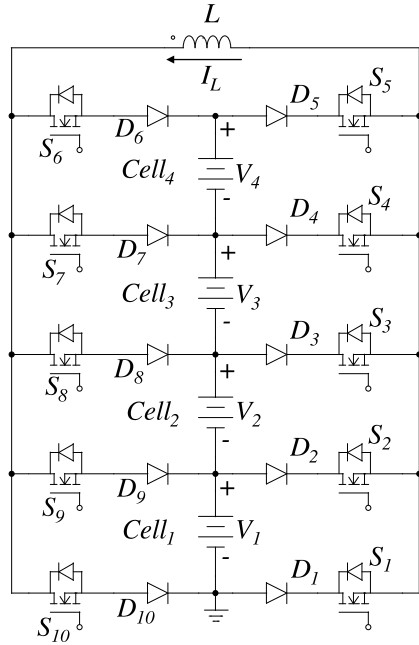

**Figure 22.** Single-inductor equalizer topology.

Figure 23 shows the simulation for a four-cell battery bank with the single inductor scheme. $Cell_2$ delivers energy to $Cell_1$, while $Cell_3$ and $Cell_4$ remain untouched. This behavior is explained because they are within an allowable range with the average voltage of the pack. Moreover, it reflects an advantage of the topologies that only use one storage device since direct C2C transfer is used for the initial conditions of the battery. Hence, in case that the battery bank is completely equalized except for two physically distant cells, topologies with one storage device perform better than other schemes (lower equalization time and better efficiency) if a direct C2C is implemented. Figure 24 shows a simulation where more than one transfer is required. In this case, the topology does not perform well, since it is handled one transfer at a time. For this simulation, an inductor of 500 μF was used, leading to a ripple of 0.4 A. Table 4 shows a comparison of the inductor-based equalizers.

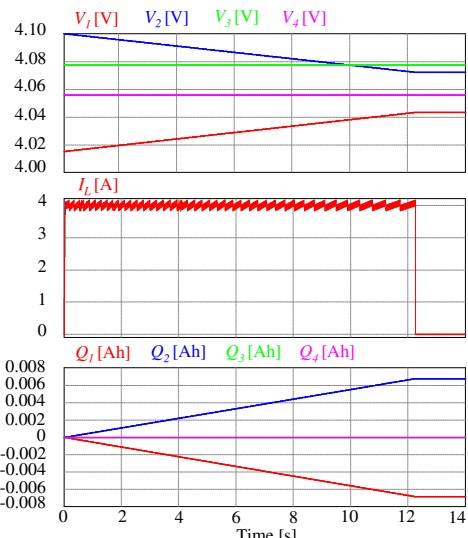

**Figure 23.** Equalization of a four-cell battery bank using the single inductor scheme.

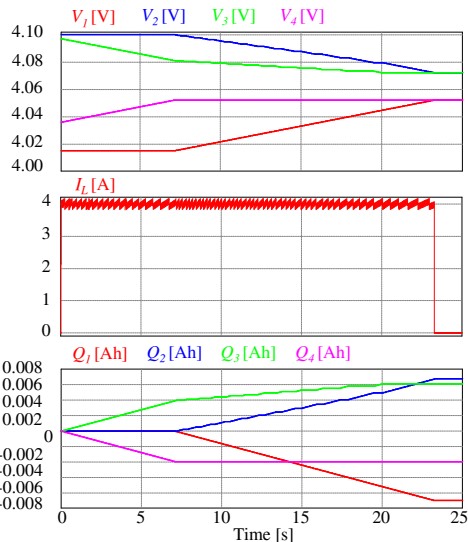

**Figure 24.** Equalization of a four-cell battery bank using the single inductor scheme.

**Table 4.** Comparative analysis of inductor-based equalizers.

| Equalizer | Component Count | Equalization Time | MOSFET Sstress | Efficiency |
|---|---|---|---|---|
| Switched inductor [57,59] | $2(n-1)$ MOSFETs $n-1$ inductors | Directly proportional to the energy to be equalized. In general, it is faster than the capacitor-based equalizers since the current is controlled. The switched inductor topology is faster than the single inductor scheme since there are more storage devices to handle the transfers of energy. | Voltage of the cell Current reference (ideally the maximum current allowed for the cell) | It presents conduction and switching losses, depending on the series resistance of the elements and the switching frequency. |
| Single inductor [60] | $2(n+1)$ switches $2(n+1)$ diodes 1 inductor | | Depends on the strategy (C2C, C2S, P2C, C2P and S2S) it vary from the voltage of the cell to the voltage of the pack Current reference (ideally the maximum current allowed for the cell) | It presents conduction and switching losses, depending on the series resistance of the elements and the switching frequency. |

### 2.2.3. Converter-Based Equalizers

Power converters have been applied to the equalization of cells in a battery pack. The main advantage of these equalizers is the control of the demanded current and the current delivered for each cell. Therefore, neither cell present a pulsating current. However, they are expensive and complex to design and to implement. Moreover, an intelligent controller is necessary for its proper operation [38,39,61].

Figure 25 illustrates an equalizer based on the bidirectional Ćuk converter. The Ćuk converter is a DC-DC non-isolated converter with the characteristic to present an inverting output voltage. It requires two inductors, one internal capacitor and two switches for every pair of adjacent cells to achieve the energy transfer. The presence of the inductors in the input and the output stage of the converter allows a smooth regulated current in both cells involved in the transfer. However, since it is needed one converter for each pair of adjacent cells, it is an expensive equalizer. This converter can be analyzed as a boost-buck converter in cascade. The source cell transfers the energy to the internal capacitor increasing the voltage, while the second stage emulates a buck converter delivering the energy to the sink cell [62,63]. The controller required is very similar to the switched inductor topology. However, the sliding surface is more complex since it is formed with the voltage of the internal capacitor and each current of both inductors. Figure 26 depicts the controller implemented for the Ćuk converter that equalize $Cell_1$ and $Cell_2$.

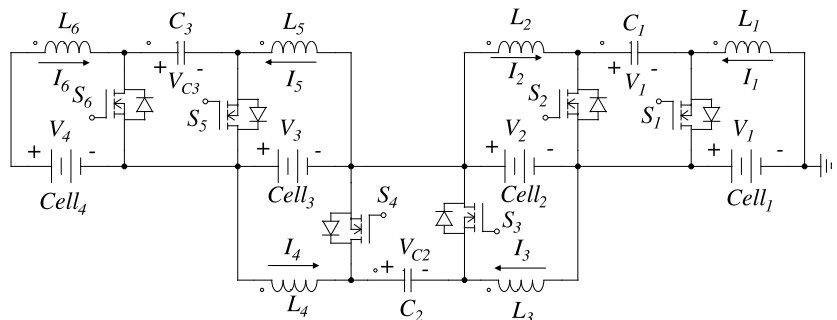

**Figure 25.** Ćuk converter equalizer.

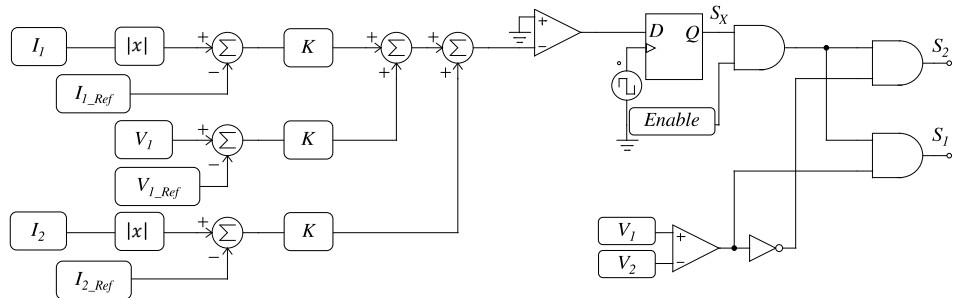

**Figure 26.** Controller required for the Ćuk converter equalizer.

Figure 27 shows the simulation for a four-cell battery bank using Ćuk converters to equalize adjacent cells. The controller used leads to a current ripple of 1 A. It is necessary to take into account this ripple when selecting the current reference to protect the cell. In this case, it was set in 1 A, which is far below the 4 A that allows the battery cell. Therefore, this process is slower than other topologies that regulate the current to a value near 4 A. The ripple obtained in this simulation was using two inductors of 500 μF. The main advantage of this scheme is the non-pulsating current in both cells. The stop condition for the equalization process in this simulation was 0.01 V. Moreover, when the equalization finishes, a sinusoidal current is present due to a resonant tank. This current passes through the cells,

which is an undesirable characteristic. The cells can be disconnected from the converter after the battery bank is equalized, but this solution requires more devices.

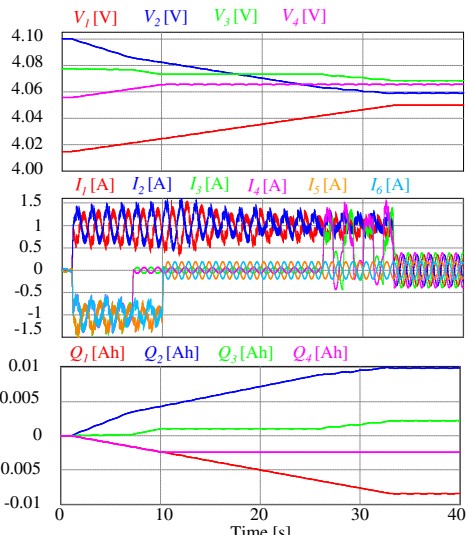

**Figure 27.** Equalization of a four-cell battery bank using the Ćuk converter topoloy.

Figure 28 depicts the switched capacitor–inductor scheme. This topology uses two storage devices to achieve the transfer. It presents a similar scheme to the switched-capacitor equalizer but is added one inductor in series with the capacitor. Therefore, there are two MOSFETs for each cell and a network formed by a capacitor and an inductor for each pair of adjacent cells. The operating principle is similar to equalize a couple of neighboring cells. First, the MOSFETs are activated to force the cell with the highest voltage to send energy to the storage elements until the capacitor reaches the voltage of the cell. The MOSFETs are then switched to send the energy storage in the previous stage to the cell with the lowest voltage. This scheme solves a disadvantage of the switched-capacitor equalizer since the inductor opposes sudden changes in current; therefore, current peaks are avoided when switching the MOSFET [64–66].

The controller is simple and straightforward since it only requires a PWM signal applied to the MOSFETs, as shown in Figure 28. Therefore, the design stage is greatly simplified. However, the current is not controlled, so the equalization time is slow. Furthermore, an analysis is necessary for the selection of the switching frequency. The goal is for the current to be zero the moment the switch changes state. In this way, the power lost during the switching of the devices is negligible. The Equation (3) determines the damped natural frequency of the system [67,68]. Where $R$ is the resistance of the circuit, $L$ is the inductance and $C$ is the capacitance. In this way, it is known that the current and the voltage numerically become zero with that frequency. Therefore, to achieve a current of zero when the MOSFETs are switched, it is sufficient to select a divisor of the frequency obtained by Equation (3).

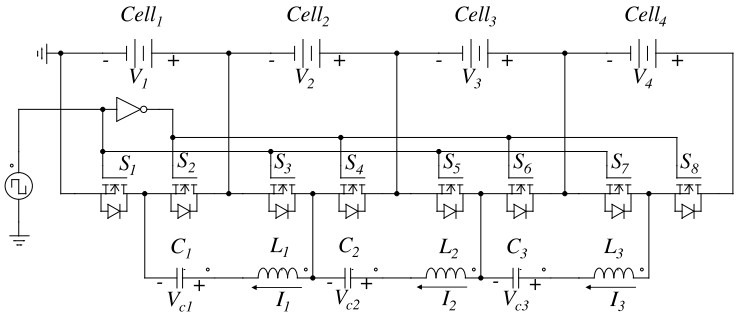

**Figure 28.** Switched capacitor–inductor network equalizer.

$$f_d(t) = \frac{1}{2\pi}\sqrt{\frac{4L - R^2C}{4L^2C}}$$ (3)

Figure 29 shows the simulation of the equalization process for a four-cell battery bank using a switched capacitor–inductor network equalizer. For these simulations, we selected an inductance value of 5 μH, a capacitance of 10 μF, and a resistance of 100 mΩ. This scheme leads to a slower process than that resulting from using the switched-capacitor scheme. Figure 30 shows a zoom of the currents and voltages of interest. Sudden changes in the current are avoided, and this is what leads to a slower process. After 40,000 s (more than 11 h), the voltage between the cells is 10 mV. Despite the large equalization time, the great advantage of this equalizer is the simplicity of its controller. A PWM signal is enough to achieve the equalization between cells. A stop condition is not even necessary, since the smaller the voltage difference between the cells, the lower the consumption of the equalizer circuit. A disadvantage of this scheme is that it presents an over-equalization in the process. Although no evidence was found in the literature, the effect of using multiple capacitor–inductor networks was analyzed. This analysis is similar to the analysis to use the multiple-tiered switched capacitor scheme. Figure 31 shows the simulations for this topology with multiple capacitor–inductor networks. In the double-tiered scheme, the equalization finishes in the 27.9% of the time, while in the triple-tiered scheme, the equalization finishes in the 13.8% of the time.

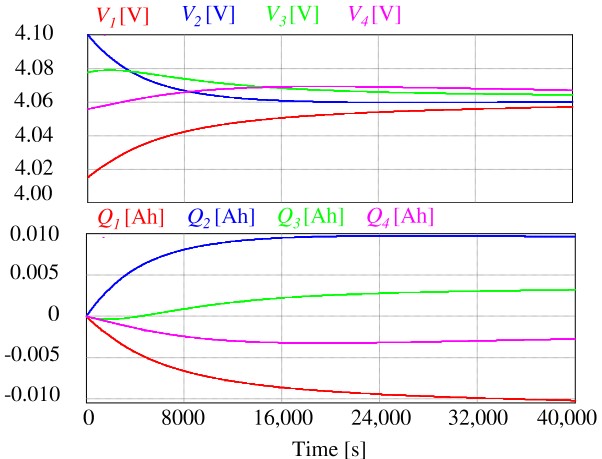

**Figure 29.** Equalization of a four-cell battery bank using the switched capacitor–inductor network scheme.

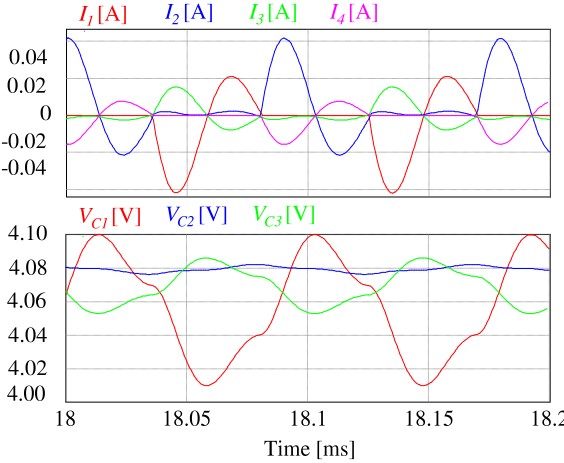

**Figure 30.** Zoom of the current in the cells and voltage of the capacitors in the switched capacitor–inductor network scheme.

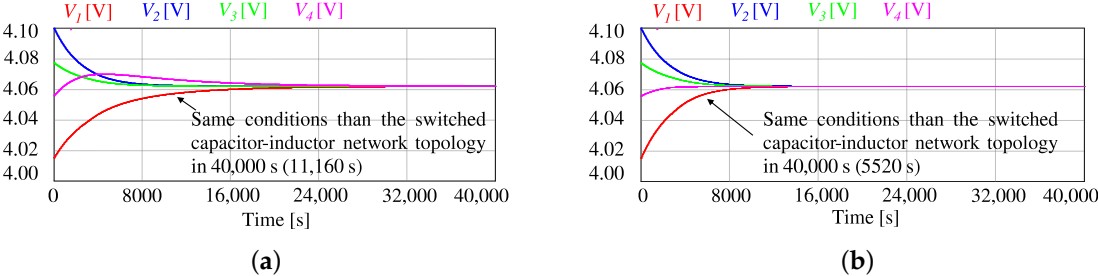

**Figure 31.** Multiple-tiered switched capacitor–inductor network topology: (**a**) double-tiered equalizer; (**b**) triple-tiered equalizer.

Figure 32 illustrates the equalizer based on buck-boost converter proposed in [69,70]. *Cell₁*, *Cell₂* and *Cell₃* transfer their exceed of energy to the cells that are physically arranged above it in the battery bank. For example, to extract power from the *Cell₂*, all the switches are kept off except the switch $S_2$. In this way, when the switch is activated, the energy is transferred from *Cell₂* to the inductor $L_2$. On the contrary, when the switch is turned off, the energy is transferred to *Cell₃* and *Cell₄* through the diode $D_2$. This equalizer requires $n$ inductors, $n$ MOSFETs and $n$ diodes to equalize $n$ cells. The circuit does not require capacitors, which are elements with a short lifespan compared to other electronic devices. However, it is an expensive scheme due to the number of inductors required. A high-level controller is also needed, as it is necessary to decide the best possible transfer. Figure 33 shows the flow diagram of the controller used in this work.

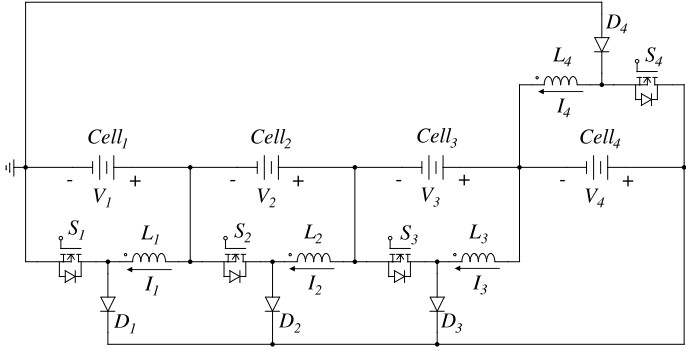

**Figure 32.** Buck-boost converter-based equalizer.

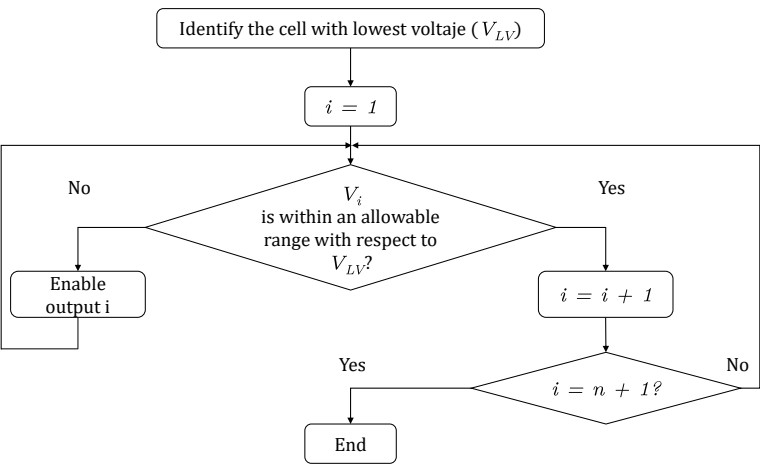

**Figure 33.** Algorithm for the operation of the buck-boost equalizer.

Figure 34 shows the simulation of the equalization process for a four-cell battery bank using an equalizer based on the buck-boost converter. For this simulation, a 200 μH inductor was used, and the current was controlled to 4 A. However, it is a slower process compared to other equalizers where the current is also controlled. This behavior is explained due to the over-equalization that exists in this scheme. The behavior of the tension in $Cell_2$, $Cell_3$ and $Cell_4$ show this process since they do not follow a straight path to their final value. This behavior can be avoided with a better algorithm in the controller, but it would increase the complexity and design time. Table 5 shows a comparison between the converter-based equalizers.

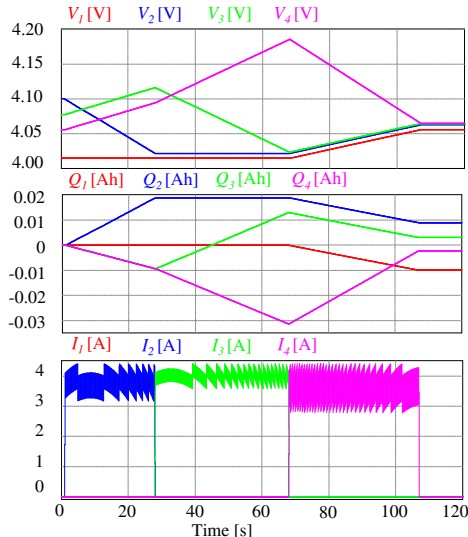

**Figure 34.** Equalization of a four-cell battery bank using the buck-boost converter topoloy.

**Table 5.** Comparative analysis of converter-based equalizers.

| Equalizer | Component Count | Equalization Time | MOSFET Stress | Efficiency |
|---|---|---|---|---|
| Ćuk converter [62,63] | $2(n-1)$ MOSFETs $2(n-1)$ inductors $n-1$ capacitors | Directly proportional to the energy to be equalized. In general is faster than the capacitor-based equalizers, since the current is controlled. The switched inductor topology is faster than the single inductor scheme since there are more storage devices to handle the transfers of energy. | $V_{cap} - V_{cell}$ Current reference (ideally the maximum current allowed for the cell) | It presents conduction and switching losses, depending on the series resistance of the elements and the switching frequency. |
| Switched capacitor–inductor network equalizer [66] | $2n$ MOSFETs $n-1$ inductors $n-1$ capacitors | | Voltage of the cell Resonance current | Only presents conduction losses, if the switching frequency is properly selected. The efficiency is affected negatively by the redundant equalization. |
| Buck-boost converter [69,70] | $n$ MOSFETs $n$ diodes $n$ inductors | | Voltage of the cell Current reference (ideally the maximum current allowed for the cell) | It presents conduction and switching losses, depending on the series resistance of the elements and the switching frequency. |

## 3. Discussion

Battery equalizers are a crucial component to ensure a safety operation in a battery bank. The balancing efficiency is an essential parameter in equalizers since the less power it consumes, the more energy transferred into the cell. In this aspect, passive methods present a poor performance when compared to active ones. Moreover, switched capacitor–inductor network equalizers and capacitor-based equalizers suppress the switching losses; hence, these equalizers offer good efficiency. The other active equalizers present switching and conduction losses; therefore, they present a lower efficiency [71].

Reference [31] discusses other factors that impact in the efficiency. The equalization variable used is crucial since the operating voltage leads to an inefficient process. This behavior is explained because the variable does not reflect the internal state of the cell. Thereupon, the equalization process will be over-activated. Moreover, the equalization strategy can also lead to repeated equalization, e.g., rationalize the equalization variable to a threshold, minimize the equalization time and maximize the battery capacity. A recommended strategy to avoid an inefficient process is to minimize energy consumption. However, it is difficult to obtain the proper data and increases the cost of the hardware needed [31,72].

Battery equalizers are a crucial part of the storage system of EVs. They take active measures to keep all cells within an allowed range of the equalization variable, even when they present a high dispersion in capacity and internal resistance [16,39,40]. In this way, the batteries are protected, which is the most expensive element in EVs. Further investigations in this area are needed to overcome the shortcomings of the reviewed topologies. Advancements need to be made to improve one or more of the critical parameters highlighted as the component count, power losses, equalization time, controller and implementation complexity, current and voltage stress in the switches, size and cost. The advantages and limitations of the topologies present in the literature were highlighted in this work. We think that this paper serves as a guideline for future research and investigations regarding the issues and challenges of this topic.

Table 6 summarizes the results obtained in the simulations of this work. The complexity of the low-level controller indicates the number of variables that need to be kept regulated. For example, in the Ćuk converter, it is necessary to control 3 variables, the current in both inductors and the voltage in the internal capacitor. The complexity in the high-level controller indicates if it only decides the stop condition (1) or if it also decides the cells for power transfer (2). Finally, for efficiency, a score of 0 was given to passive schemes, 1 to those that present switching and conduction losses and 2 to equalizers that only present conduction losses.

Metrics from Table 6 were used to compare the reviewed equlizaers with an idea equalizer. An ideal equalizer has few devices, low equalization time, low switch stress, low controller complexity and high efficiency. According to their approximation to the ideal equalizer, we assign them a number where 1 is the most desirable equalizer. For example, the shunt MOSFET equalizer is 1 in component count because it is the equalizer with the fewest components. All the places assigned in the previous step for a converter are added to take into account all the parameters. The last column of Table 6 shows the accumulated places for each converter. Finally they are ordered from lowest to highest, where the converter with the lowest score is the one with the most desirable characteristics. The best equalizer using this methodology is the switched capacitor. However, this procedure is quite simple and has many points that can be improved. For example, weighted coeficients can be used to highlight parameters of interest. In this way, using this table a designer can reach important conclusions to select an equalizer suitable for his application.

**Table 6.** Comparative analysis of the reviewed equalizers for a four-cell battery bank.

| Equalizer | Component Count | Equalization Time [s] | MOSFET Stress | Low-Level Controller Complexity | High-Level Controller Complexity | Sensors Rrequired | Efficiency | Total |
|---|---|---|---|---|---|---|---|---|
| Switched resistor [40] | 4 resistors, 4 MOSFETs-(2) | 17.7-(2) | 4.1 V, 4.1 A-(6) | 0-(1) | 1-(4) | 4 (V)-(4) | 0-(9) | 28 |
| Shunt MOSFET [44] | 4 MOSFETs-(1) | 18.18-(3) | 4.1 V, 4.05 A-(5) | 0-(1) | 1 (4) | 4 (V), 4 (A)-(8) | 0-(9) | 31 |
| Switched capacitor [51,52] | 8 MOSFETs, 3 capacitors-(4) | 8000-(8) | 4.1 V, 0.015 A-(1) | 0-(1) | 0-(1) | 0-(1) | 2-(1) | 17 |
| Single capacitor [49,50] | 16 MOSFETs, 1 capacitor-(9) | 28,000-(9) | 4.1 V, 0.015 A-(1) | 0-(1) | 2-(8) | 4 (V)-(4) | 2-(1) | 33 |
| Double-tiered capacitor [54] | 8 MOSFETs, 5 capacitors-(6) | 3200-(7) | 8.2 V, 0.23 A-(10) | 0-(1) | 0-(1) | 0-(1) | 2-(1) | 27 |
| Switched inductor [57,59] | 6 MOSFETs, 3 inductors-(3) | 16-(1) | 4.1 V, 4.4 A-(7) | 1-(7) | 1-(4) | 4 (V), 3 (A)-(7) | 1-(5) | 34 |
| Single inductor [60] | 10 switches, 10 diodes 1 inductor-(10) | 23-(4) | 4.1 V, 4.4 A-(7) | 1-(7) | 2-(8) | 4 (V), 1 (A)-(6) | 1-(5) | 47 |
| Ćuk converter [62,63] | 6 MOSFETs, 6 inductors 3 capacitors-(8) | 33, (5) | 4.1 V, 1.5 A-(4) | 3-(10) | 1-(4) | 7 (V), 6 (A)-(10) | 1-(5) | 46 |
| Switched capacitor–inductor network equalizer [66] | 8 MOSFETs, 3 inductors 3 capacitors-(7) | 40,000-(10) | 4.1 V, 0.09 A-(3) | 0-(1) | 0-(1) | 0-(1) | 2-(1) | 24 |
| Buck-boost converter [69,70] | 4 MOSFETs 4 diodes 4 inductors-(5) | 108-(6) | 4.1 V, 4.4 A-(7) | 1-(7) | 2-(8) | 4 (V), 4 (A)-(8) | 1-(5) | 46 |

*Future Trends*

Several topics offer promising opportunities to improve existing BECs. In recent years, reconfigurable battery systems were proposed to achieve battery equalization. Reconfigurable batteries define their connections between cells by software. The battery reconfiguration allows the connection of cells/modules to be flexible and adapt the battery pack to the charging/discharging requirements. In cell equalization, this feature is used to send the larger current to the lower SOC batteries. Commonly,two to six switches are used per cell to obtain a flexible battery pack [73,74]. A drawback of this method is that the reconfiguration must be offline to prevent dangerous situations. If this strategy is combined with the equalizers reviewed, the balance performance could be improved and even optimized [73].

Moreover, current equalizers present a fixed volume of equalization depending on the battery pack. The component count of the BEC relies on the number of cells of the battery pack and the architecture selected. Sometimes, it is desirable to design a circuit with the components to use before-hand. In BECs, that challenge has not been widely studied [75,76].

There is also interest in the integration of the BECs with the charger of the vehicle. In general, the trend in the power converters applied to EVs is the integration to achieve more functions with fewer electronics. It is possible to use one circuit to realize both functions, the charging process and the equalization [9]. However, this technique can only be applied in level 1 or 2 of charging since the fast charging is with an off-board converter. This method is a challenge for future investigations and has the opportunity to create a cost-effective converter when compared to dedicated converters for each function [40,77].

The high-level controller is another subsystem that offers promising research opportunities. In contrast with the BECs the investigation on control strategies is further behind. The SOC and capacity estimators need to be more stable and accurate without requiring powerful real-time implementation hardware. Moreover, the stability and accuracy of estimators through the whole lifecycle of the cells is a major challenge. The equalization objectives need to be designed in a multi-objective perspective instead of using a singular objective approach. However, all technical indexes are difficult to meet; therefore, to set the proper constraints for the indexes remains unsolved. Finally, any classic, intelligent or data-driven controller requires a model for the battery pack and, since the cells are complex nonlinear time-varying systems, this area needs further investigations [31,78,79].

Another trend in EVs that can play a significant role in battery equalizers is the wireless power transfer. This technology does not require heavy cables, connectors, etc. Therefore, it can be achieved a low cost and light-weight system with great flexibility and reliability. Moreover, it can make obsolete the high-level controller since the cells will have the possibility to communicate and make decisions. This groundbreaking technology will have a huge impact on battery equalization, charge and other applications offering a promising field of study in the coming years [80,81].

## 4. Conclusions

The main parameters to select in an EV are its range, charging time, and acceleration. All these elements are related to the battery of the vehicle. Moreover, this device represents the most expensive, heaviest and bulkiest component in the system. Therefore, it is imperative to protect this element against dangerous situations. The main challenge in a battery pack used in EVs is to keep all cells within an allowable range of SOC or voltage operation. The array of cells connected in series to form a battery pack are charged/discharged at a different rate. This behavior is due to the dispersion in internal capacity, resistance, self-discharge rate and uneven distribution of the temperature in the battery pack. If the battery pack is operated without a battery equalizer, it decreases the service life of the cells and can even explode. BECs prevent these situations by taking active measures to keep all the cells within the same SOC.

The equalizers are classified, taking into account the main component in its architecture and energy dissipation. Passive equalizers present a low efficiency because they burn the excess of energy

through a resistor. For this reason, it is not widely used for high-power applications. In contrast, active equalizers are preferred since the cell/string of cells with the highest SOC transfer the excess of energy to other cells/string of cells with lower SOC. The capacitor-based equalizers are simple to control and inexpensive, but present surge currents that can damage the cells. In the inductor-based equalizers, the current is controlled and they have a short equalization time. However, they present magnetization losses, saturation problems, are expensive and are bulky. The equalizers based on converters control the current extracted from the cell and the current delivered. Nevertheless, they are complex to control and expensive. The equalizer selection depends on the application, the budget and the critical parameter to consider for the designer. In this work we used a methodology to compare the main topologies based in quantitative simulation data. According to the results, the best active equalizers are the switched capacitor and the switched capacitor–inductor network.

Promising research opportunities were highlighted to move forward in this topic of investigation. Combining the existent equalizers with reconfigurable batteries could improve the equalization process. However, there are more questions than answers in the reconfiguration of the battery online. Although Tesla, Microsoft and several top tier universities accepted the software-defined batteries as a promising technology for EVs we consider that this technology will not reach its full potential in the near future. Moreover, another research opportunity is to design the volume of the equalization circuit to operate the battery pack. We think that this research trend can play a major role in the near future, due to the flexibility in the design process. Since the number of components can be accommodated to the application, it can be designed to optimize several parameters such as the cost, equalization time, number of components, etc. Finally, another identified trend is the integration of the battery equalizer and the charger, to achieve a multi-purpose converter instead of two dedicated converters. This approach has proved to be cost-effective in other applications. In general, the batteries need in-depth research since it is well known that it is the main hurdle for the widespread use of EVs.

**Author Contributions:** Conceptualization, A.A.-D., M.-A.M.-P., S.T. and J.R.-R.; methodology, R.V.C.-S., J.R.-R. and M.-A.M.-P.; software, A.A.-D., S.T. and A.A.E.-B.; validation, A.A.-D., A.A.E.-B., R.V.C.-S and M.-A.M.-P.; formal analysis, R.V.C.-S. and J.R.-R.; investigation, A.A.-D., A.A.E.-B. and S.T.; resources, J.R.-R., M.-A.M.-P. and R.V.C.-S.; data curation, M.-A.M.-P., S.T. and J.R.-R.; writing—original draft preparation and writing—review and editing, all authors. All authors have read and agreed to the published version of the manuscript.

**Funding:** This research was funded by CONACyT and PRODEP.

**Conflicts of Interest:** The authors declare no conflict of interest.

## Abbreviations

The following symbols are used in this manuscript:

| | |
|---|---|
| EV | Electric vehicle |
| ICE | Internal combustion engine |
| BEC | Battery equalizer circuit |
| Li-ion | Lithium-ion |
| USA | United States of America |
| SOC | State of charge |
| MOSFET | Metal–oxide–semiconductor field-effect transistor |
| C2H | Cell-to-heat |
| C2C | Cell-to-cell |
| C2S | Cell-to-string |
| P2C | Pack-to-cell |
| C2P | Cell-to-pack |
| S2S | String-to-string |
| PWM | Pulse width modulation |
| PHEV | Plug-in hybrid electric vehicle |

$V_x$      Voltage between two nodes. Can be from a capacitor or a voltage source.

$C_x$      Capacitor $x$ present in a topology.

$L_P$      Primary winding of a transformer.

$L_S$      Secondary winding of a transformer.

$L_x$      Inductor $x$ present in a topology.

$D_x$      Diode $x$ present in a topology.

$S_x$      MOSFET $x$ present in a topology.

$Sb_x$      Bidirectional switch $x$ present in a topology.

$I_P$      Current across the primary winding of a transformer.

$I_L$      Current across the secondary winding of a transformer.

$I_x$      Current across the inductor $x$.

$Cell_x$      Cell $x$ of the battery bank.

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
