# Peer review of "A Review of Battery Equalizer Circuits for Electric Vehicle Applications"

_energies, doi:10.3390/en13215688_

Round 1

Reviewer 1 Report

The paper was properly revised as per the comments.

Reviewer 3 Report

The paper presents a 81-item literature review of the current
state of the art regarding battery equalizer circuits for
electric vehicles. In comparison to the first version of the
manuscript authors decided to reduce total citing items from
initial 112 by 31.

Also authors decided to remove subsections describing the various
technologies of batteries which was not related to the main topic
of the work.

For the discussed battery equalizer systems, simulations were
made during which the plots of the changes in selected parameters
(voltages, charges and currents) were visualized, which
significantly improved the quality of work.

Overall readability of drawings have been corrected.

In Table 6. Authors provide interesting method to rank the
described BECs which can be useful in the BEC on the stage of
its selection to the respective application.

The paper currently addresses a very important and interesting
topic. Considering all paper and the above remarks, I recommend
accepting the work for publication in present form.

This manuscript is a resubmission of an earlier submission. The following is a list of the peer review reports and author responses from that submission.

Round 1

Reviewer 1 Report

The authors have presented a thorough review on battery equalizer circuits for electric vehicle applications. The reviwer has the following suggestions:

-The authors should clearly highlight more future directions.

-The authors' view should be clearly stated in the conclusions.

Reviewer 2 Report

The manuscript is an elaborate review paper on the battery equalizer circuits for electric vehicle. It provides useful comparative analysis of existing batteries and equalizer circuits. 

As one suggestion, however, it is hard to get important takeaway points in the discussion section. Please make specific subsections that can clarify the importance, trend, difficulty, future directions, and etc. of the reviewed technology. 

Reviewer 3 Report

The paper investigated the review of battery equalizer circuits for EV application. The related works were properly summarized in the paper but it should be clarified including the followings before considering publication.

- The performance of each type of equalizer should be compared quantitatively presenting the experimental and simulation results.
- A state of the art battery systems should be investigated and compared to the conventional ones.
- Nomenclature and abbreviation shoud be added for the better understanding.
- English should be improved throughout the manuscript.

Reviewer 4 Report

The paper presents a 112-item literature review of the current state of
the art regarding battery equalizer circuits for electric vehicles.
There are several articles on similar subject, but this article
contains a collection of solutions from recent years
which is
its biggest advantage.

In my opinion the readability of the work is impaired by the
lack of description of the symbols used in all drawings.
In addition, in Figure 17 one source is unmarked.

The number of cycles (1500 - 2000) reported for lithium-ion
cells in Table 1 should be corrected.
Depending on the type of cells and operating conditions, it
can be in the range from 1000 (NMC) to even 10,000 (LTO).

There is a typo in line 211 "Pasive".

The discussion contained in lines 392 - 417 does not apply
to the subject of the article. Please consider changing or moving this
section to the chapter on batteries.
To sum up, the paper currently addresses a very important and
interesting topic. Considering all paper and the above remarks,
I recommend accepting the work for publication after
making minor corrections.